# *TAS2R38* Bitter Taste Receptor Polymorphisms in Patients with Chronic Rhinosinusitis with Nasal Polyps Preliminary Data in Polish Population

**DOI:** 10.3390/biomedicines12010168

**Published:** 2024-01-12

**Authors:** Joanna Jeruzal-Świątecka, Edyta Marta Borkowska, Martyna Borkowska, Wioletta Pietruszewska

**Affiliations:** 1Department of Otolaryngology, Head and Neck Oncology, Medical University of Lodz, al. Tadeusza Kościuszki 4, 90-419 Lodz, Poland; joanna.jeruzal@office365.umed.pl; 2Department of Clinical Genetics, Medical University of Lodz, 90-419 Lodz, Poland; edyta.borkowska@umed.lodz.pl (E.M.B.); martyna.borkowska2@stud.umed.lodz.pl (M.B.)

**Keywords:** bitter taste receptor, *TASR2R38*, polymorphisms, chronic rhinosinusitis, nasal polyps

## Abstract

Chronic rhinosinusitis (CRS) affects 5–12% of the general population, and the most challenging patients are those with nasal polyposis (CRSwNP). Its complexity, unpredictability, and difficulties in selecting a treatment plan individually for each patient prompted scientists to look for possible genetic causes of this disease. It was proven that single nucleotide polymorphisms (SNPs) in the *TAS2R38* gene may affect the mobility and the activity of the ciliated epithelium of the upper respiratory tract what can contribute to individual differences in susceptibility to CRS. There are two common haplotypes: a “protective” type (PAV), and a “non-protective” type (AVI). CRS patients who are homozygous PAV/PAV are considered as less susceptible to the severe course of the disease, whereas patients with AVI/AVI haplotype are more vulnerable. The aim of this study was to examine *TAS2R38* gene polymorphisms among CRSwNP patients and control group (N = 544) with the evaluation of the association between the distribution of studied polymorphic variants and the incidence as well as severity of CRSwNP in the study group. Whole blood samples from CRSwNP patients (N = 106) and the control group (N = 438) were analyzed for alleles of the *TAS2R38* gene using real-time PCR single nucleotide polymorphism genotyping assays for rs713598, rs1726866, and rs10246939. PAV (SG: 41%; CG: 49%) and AVI (SG: 59%; CG: 51%) haplotypes were the only ones detected in the study. The AVI haplotypes were 1.5 times more frequent in the study group than in the control group (*p* = 0.0204; OR = 1.43). AVI/AVI individuals tended to have more severe symptoms in the VAS scale, less QoL in the SNOT-22 test, and a bigger nasal obstruction upon endoscopic examination. Patients with PAV/PAV were twice more likely to have minor changes in preoperative CT scans (*p* = 0.0158; OR = 2.1; Fi = 0.24). Our study confirmed that the PAV/PAV diplotype might have some protective properties and carrying the AVI haplotype might predispose to the development of CRSwNP.

## 1. Introduction

Chronic rhinosinusitis (CRS) is a significant health problem that affects 5–12% of the general population [1]. Its diagnosis requires a duration of at least 12 weeks and the presence of at least two of the following symptoms: nasal blockage/nasal obstruction/nasal congestion or nasal discharge (anterior/posterior nasal drip) and facial pain/pressure or the reduction/loss of smell. There are two major documents that organize the knowledge on CRS, its diagnosis, classification, and treatment: the European Position Paper on Rhinosinusitis and Nasal Polyps (EPOS) with its update from 2020, and the International Consensus Statement on Allergy and Rhinology: Rhinosinusitis (ICAR-RS) from 2021 [1,2]. ICAR-RS kept the existing classification of CRS into chronic rhinosinusitis with (CRSwNP) and without (CRSsNP) nasal polyps, based on the presence or absence of nasal polyps. However, a new classification of CRS into primary and secondary CRS and further division into localized and diffuse disease, based on anatomic distribution, was proposed in EPOS 2020. In primary CRS, the disease is classified based on endotype dominance, either Type 2 or non-Type 2, referring to the type of immunological response that causes the inflammation. In this categorization, CRSwNP is a primary, diffused, Type-2 disease. These visible differences, in the form of CRS categorization, illustrate the complexity of the causality and etiology of this disease. The search for genetic causes of CRS has become one of the pathways to better understand this condition. Currently, the only fully proven genetic cause of CRS is a genetically determined ciliary impairment, as seen in patients with Kartagener’s syndrome and primary ciliary dyskinesia (PCD), as well as nasal mucus dysfunction in cystic fibrosis (CF). Recent studies focusing on genetic and epigenetic aspects of nasal polyposis link this medical condition with genes involved in inflammation and immune response, cytokine genes, leukotriene metabolism, and the extracellular matrix [3]. Some of them have shown that abnormalities in the mobility and the activity of the ciliated epithelium of the upper respiratory tract may be related to the bitter taste receptors (T2Rs): *TAS2R13*, *TAS2R19*, *TAS2R38*, and *TAS2R49* [4,5], out of which, *TAS2R38* seems to have major significance [6,7,8,9]. This receptor plays an important role in the detection of microbes and, when activated, produces a calcium-dependent increase in nitric oxide (NO) production and ciliary beat frequency (CBF). This results in both: an increased mucociliary clearance (MCC) and an augmented innate antimicrobial effect [6]. It was established that single nucleotide polymorphisms (SNPs) in the *TAS2R38* gene may have an impact on individual differences in vulnerability to respiratory infections, particularly to CRS [5,10]. The effectiveness of this immune defense depends on the three most common polymorphisms. Those variants are associated with amino acid located at Positions 49 (rs713598; Ala49Pro) encoding alanine or proline, 262 (rs1726866; Ala262Val) encoding alanine or valine, and 296 (rs10246939; Ile296Val) encoding isoleucine or valine. These three polymorphisms are interconnected and generate two common haplotypes. In the “protective” type, the *TAS2R38* allele codes proline (P), alanine (A), and valine (V), creating the PAV haplotype. In the “non-protective” type, the receptor allele codes alanine (A), valine (V), and isoleucine (I) create the AVI haplotype. Patients who are homozygous PAV/PAV are considered “supertasters” and AVI/AVI “non-tasters”, while the heterozygous PAV/AVI show a wide range of bitter taste receptions [11]. “Supertasters” seem less likely to develop chronic rhinosinusitis and, in case of the disease, have less severe symptoms and fever recurrence. CRS patients with AVI/AVI genotypes seem more susceptible to a severe course of the disease and more often require surgical treatment [12,13,14]. Although the remaining haplotypes (AVV, AAI, AAV, PAI, PVV, and PVI) and their combinations have been very rarely observed, some of them constitute a significant part of certain specific populations [15,16]. So far, the Polish population of CRS patients has been little studied in this regard. Dżaman et al. enrolled 20 CRS patients in their study and found that the frequency of heterozygotes (AVI/PAV) was the highest and that the protective genotype (PAV/PAV) occurred with the lowest frequency and related to a lower average value of CT score compared to AVI/AVI genotypes (*p* = 0.01) [17].

The aim of this study was to identify the *TAS2R38* bitter taste receptor polymorphism in CRSwNP patients in comparison to the control group and confront the haplotype distribution with clinical features of CRSwNP. 

## 2. Materials and Methods

The research was conducted from December 2017 until May 2023 and included 544 individuals. The study group consisted of 106 patients with CRSwNP and the control group was formed by 438 volunteers. All participants signed a voluntary, informed consent to the laryngological examination, blood sample collection, and further genetic tests. The study was approved by the Bioethics Committee of the Medical University of Lodz (RNN/05/18/KE).

### 2.1. Patients’ Enrolment and Clinical Data Collection

In this case-control study, the inclusion criteria for the study group were defined as follows: adults of Caucasian origin (based on physical features and family history) qualified for primary functional endoscopic sinus surgery (FESS) for CRSwNP and operated in the Department of Otolaryngology, Head and Neck Oncology, Medical University of Lodz, The Norbert Barlicki Memorial Teaching Hospital, Lodz, Poland. Adult patients and medical students of Caucasian origin (based on physical features and family history) without a history of chronic or acute rhinosinusitis who agreed to participate in the study by providing a blood sample formed the control group. Patients with known autoimmune dysfunction, immune deficiency, primary ciliary dyskinesia, cystic fibrosis, any history of radiation exposure to the paranasal sinuses, and oncological treatment were excluded from both groups.

All patients were tested for phenylthiocarbamide (PTC) tasting capability by administrating two drops of PTC 0.025% aqueous solution on the tongue (Kolchem, Lodz, Poland). PTC was chosen since it is a ligand for *TAS2R38*. Bitter taste perception after the administration was defined as a positive result, and lack of bitter taste sensation was defined as a negative result [18,19,20].

CRS patients filled out a questionnaire concerning data such as age, height, weight, patient’s age at the time of first symptoms, and the severity of clinical symptoms according to the Visual Analog Scale (VAS) and Sino-nasal Outcome Test-22 (SNOT-22), along with information about asthma, chronic obstructive pulmonary disease (COPD), and allergy. Symptoms such as nasal blockage/obstruction/congestion or nasal discharge (anterior/posterior nasal drip), facial pain/pressure, and reduction or loss of smell were rated by patients on a VAS scale from 0 to 10 where “0” signified no symptom presence and “10” indicated its greatest severity. This score, evaluated before the surgery, was divided into three categories based on the results of a validation study. Mild disease was defined as a VAS score of 0–3, moderate as >3–7, and severe as ≥7. SNOT-22 test was completed by each patient by answering all 22 questions on a 0–5 scale, where “0” meant no problems with a particular symptom and “5” defined the highest level of discomfort [21].

The severity of the disease was established based on endoscopic nasal examination and preoperative computed tomography (CT). Endoscopic examination of the nostrils was rated in a three-point classification system (0: absence of polyps; 1: polyps in middle meatus only; 2: polyps beyond middle meatus but not blocking the nose completely; 3: polyps completely obstructing the nose), according to Lund–Kennedy scale [22]. The Lund–Mackay score was used for measuring the obstruction of the sinuses in CT scans conducted before the surgery. According to scale: 0 = no abnormalities, 1 = partial opacification, 2 = total opacification, for all sinus systems, except the ostiomeatal complex where 0 means not occlusion and 2 = occlusion. A total score of 0 to 24 was possible, and each side is considered separately (0–12) [23].

### 2.2. DNA Genotyping for TAS2R38 Polymorphism

DNA isolation was performed with Maxwell^®^ RSC Blood DNA Kit (Promega, Madison, WI, USA) and the material was stored in a −20 Celsius degree freezer in the Department of Clinical Genetics of Medical University of Lodz. Its quantity and purity were checked by spectrophotometric measurement (OD 260/230 nm indicators, OD 260/280 nm). We genotyped alleles of the *TAS2R38* gene using real-time PCR single nucleotide polymorphism genotyping assays for rs713598, rs1726866, and rs10246939 (C_8876467_10, C_9506827_10, and C_9506826_10; Thermo Fisher, Pleasanton, CA, USA, cat No 4351379) and TaqMan™ Genotyping Master Mix (Thermo Fisher, Vilnus, Lithuania) with BioRad CFX 96 (BioRad Laboratories Inc., Hercules, CA, USA) [24,25]. No template control (NTC) was included in all analyses. All the results and allelic discrimination were conducted manually according to the detailed scheme provided by Thermo Fisher for each assay and additionally analyzed with BioRad CFX Maestro Software 2.3 (BioRad Laboratories Inc., Hercules, CA, USA) [26,27,28].

### 2.3. Statistical Analyses

Continuous variables were summarized using means and standard deviations or counts and proportions. The Chi-square test or, depending on the assumptions met, tests from this family (V-square, Chi-square with Yates correction) were used to test the relationship. The significance level was set at 5%. Odd ratios and 95% confidence intervals (CI) were calculated and reported if the relationship was significant. Statistical analysis was conducted using Statistica Software Version 13.3 (StatSoft, Krakow, Poland). A flowchart presented on Figure 1 illustrates all stages of the study.

## 3. Results

Genotyping data were obtained from 438 volunteers who formed the control group and 106 CRSwNP patients from the study group. The groups were similar in terms of age and sex with a slight majority of women in the control group. There were 62 males (58.49%) and 44 females (41.51%) in the study group. The control group consisted of 161 males (36.76%) and 277 females (63.24%). The mean age in both groups was 46 years. More than 65% of patients from the study group felt the bitter taste of PTC. Less than 30% of them admitted seasonal or year-round allergies (23.58%; 25.47%). The median SNOT-22 score was 32 (SD: 9.037), the median score in Lund–Mackey was 9 (SD: 5.474), and the median score in Lund–Kennedy endoscopic examination scale was 3 (SD: 1.14). More than half of the patients marked their symptoms as “4” or more on the VAS scale (55.66%), and for most of them, the symptoms began between 16 and 40 years of age (48.11%) (Table 1).

By genotyping whole blood samples, we determined the three *TAS2R38* receptor gene variants (rs713598, rs1726866, rs10246939) in the study group and control group. We analyzed all PCR reactions for all three polymorphisms according to the assay’s instructions provided by Thermo Fisher. A positive control was included with each polymorphic variant. Schematic allelic discrimination for all three polymorphic variants is shown in Figure 2 for one of the study group patients.

The haplotypes PAV and AVI were the only haplotypes determined in the study. The frequency of the observed haplotypes of the *TAS2R38* bitter taste gene polymorphism was not significantly different from the Hardy–Weinberg distribution (chi^2^, 1 degree of freedom, HW-E *p* = 0.0528). We detected “3” from the possible 28 diplotypes in both studies and control groups. In the control group, the PAV/AVI genotype was the most frequent (53.5%), followed by AVI/AVI (23.7%) and PAV/PAV (22.8%). The distribution of examined haplotypes in the study group was similar. However, the PAV/PAV diplotype was significantly less frequent (PAV/AVI = 56.6%, AVI/AVI = 31.1%, PAV/PAV = 12.3%; *p* = 0.01). Despite the slightly higher prevalence of the AVI/AVI homozygote in the control group compared to PAV/PAV homozygotes, PAV/PAV was chosen as the reference genotype because PAV haplotype is more frequent in the global population [27]. Heterozygotes PAV/AVI, the most common in the general population, were also the most frequent in our study both in the study and the control group. The possibility of PAV/AVI diplotypes in the study group was 1.97 times higher than PAV/PAV diplotypes (OR = 1.97; *p* = 0.0360). AVI haplotype carriers were twice as frequent in the study group (OR = 2.11; *p* = 0.01611). Also, AVI/AVI homozygotes were twice as frequent in the study group compared to the control group (OR = 2.44; *p* = 0.01). By analyzing the probability of the occurrence of individual haplotypes on each chromosome, we showed that the rates of AVI haplotype were almost 1.5 times higher in the study group than in the control group (OR = 1.43; *p* = 0.0204). That means that carrying the AVI haplotype might increase the risk of developing CRSwNP (Table 2). The distribution of diplotypes is shown in Figure 3.

Following the genetic analysis, we examined the relations between diplotypes and clinical features. Patients with PAV/PAV are twice as likely to have minor changes in preoperative CT scans (*p* = 0.0158; OR = 2.1; Fi = 0.24). Those patients seem to be more likely to develop a year-long allergy (*p* = 0.0431; OR = 4.2; Fi = 0.24). We did not observe any other statistically significant associations between individual polymorphisms and clinical features. However, some trends were possible to notice (Table 3). In the VAS scale, the AVI/AVI patients scored 8–10 points much more frequently than PAV/PAV individuals. The same tendency can be noticed in SNOT-22 and Lund–Kennedy scales. The AVI/AVI patients seemed to have scored over the median in booth scales, while the PAV/PAV individuals were more likely to have scores under the median. 

## 4. Discussion

Our study is the first one to be conducted in the Polish population in such a large group (N = 544). We observed that the PAV/PAV individuals, who are “supertasters”, had less advanced inflammation of sinuses, which might suggest some protective properties of this diplotype. Our study confirms the earlier findings in this area, which were conducted on smaller groups [29].

Despite the objectivity of genome research, particular polymorphisms of particular genes may vary significantly in different populations. Polymorphisms of the *TAS2R38* bitter taste receptor gene were studied in many different populations and fields. 

According to Risso et al., the PAV and AVI haplotypes are predominant (50.76% and 42.70%, respectively) in the global frequency of the *TAS2R38*, followed by AAI (3.39%) and AAV (2.48%) [30]. Other haplotypes occur at very low frequencies: AVV (0.32%), PAI (0.18%), PVV (0.10%), and PVI (0.07%). Although the AAI haplotype is more commonly present in Africa (7–33%), it also exists in the European population: PAV = 45.66%, AVI = 49.22%, AAV = 3.56%, AVV = 0.49%, PAI = 0.32%, PVI = 0.03%, and AAI = 0.55%. Nevertheless, even the European populations differ from each other. Carrai et al., by studying two Caucasian populations, reported that in the German population (N = 1311), the PAV/PAV, PAV/AVI, and AVI/AVI diplotypes accounted for 14%, 37%, and 29% respectively [31]. This haplotype distribution was similar in the Czech population (N = 1224) (PAV/PAV = 15%, PAV/AVI = 38%, and AVI/AVI = 27%). In both populations, rare haplotypes (AAV; PVV; AAI; PVI and PAI) accounted for the same cumulative frequency (4%) but were differently distributed. In the study conducted on the Italian population of Sardinia Island (N = 373), the PAV/AVI heterozygotes were also the most frequent (45%) [32]. The PAV/PAV homozygotes accounted for 24%, and while AVI/AVI homozygotes accounted for 27%. But some of the rare diplotypes were also revealed: AAV/AVI = 4, PAV/AAV = 4, AAI/AVI = 3, PVI/AVI = 2, AAV/AAV = 1, and PAV/AAI = 1. In the Finnish population (N = 1903), the difference between PAV/AVI and AVI/AVI was not that significant (43.6% versus 39.5%), and across all subjects, 5.5% carried at least one AAV haplotype [16]. Furthermore, the AVV and PAI haplotypes are commonly defined as extremally rare, but in the study of Ramos–Lopez et al., they were the most predominant among the Mexican–Mestizo population (N = 375), accounting for over 96% of all haplotypes (AVV = 60%, and PAI = 36.5%) [33]. In our study, we found PAV and AVI haplotypes to be the most frequent in both groups. Due to the size of the group, we cannot directly transfer our results to the entire Polish population, but our research offers a general view of the distribution of the *TAS2R38* gene polymorphism.

The importance of particular polymorphisms of *TAS2R38* in determining the functionality of the receptor itself is still not fully understood. As reported so far, the presence of alanine at Position 49 and valine at Position 262 diminishes the receptor function, whereas the variation in Position 296 has little effect on the *TAS2R38* receptor’s activation [13]. Bufe et al. exemplified this hypothesis when they observed that the AVI and AVV variants do not respond to PTC and PROP, whereas the PAV and PAI variants respond equally strongly [24]. This was confirmed by Boxer et al. In their study group (PAV = 42.3%, AVI = 53.1%, AAV = 2.5%, AAI = 1.2%, PAI = 0.8%, and PVI = 0.1%), where the participants rated their bitter taste perception to a PROP filter disc, patients with AAI, AAV, and PAI haplotypes presented an intermediate taste sensitivity [34]. 

*TAS2R38* gene polymorphisms are associated with different diseases from obesity, through gastric and colorectal cancer, PCD, CF, and CRS, to Parkinson’s disease [12,13,14,17,31,35,36,37,38,39,40,41,42,43]. Most of the studies concerning chronic rhinosinusitis confirm the differences in *TAS2R38* polymorphism distribution between CRS patients and healthy controls, highlighting that the nonfunctional genotype (AVI/AVI) is more frequent among CRS patients than in the general population. This diplotype was considered an independent risk factor for CRS requiring FESS, a higher risk for Gram-negative infections, and biofilm formation [14,39,40,41]. Moreover, the PAV/PAV diplotype was defined as the favorable prognostic factor in terms of the quality-of-life (QoL) outcome after surgery in CRSsNP patients and connected with a lower average value of preoperative CT Lund–Mackay scores [14,17]. However, an Italian research team published results questioning this theory. Gallo et al. reported no significant difference in the distribution of polymorphisms in CRS patients and controls. They also found no significant differences in genotype distribution between CRSwNP and CRSsNP, nor any significant correlation between any CRS-related risk factors, clinical features, or a particular genotype in their study group [42]. Our results suggest that PAV/PAV homozygotes have less advanced inflammatory changes visualized in the preoperational CT scans, while AVI/AVI patients had more advanced symptoms in the VAS scale, SNOT-22 questionary, and Lund–Kennedy scale. This might suggest some protective properties of the PAV/PAV diplotype. The AVI haplotype was significantly more frequent in the study group compared to the control group, indicating that this haplotype carrier may be predisposed to the development of chronic sinusitis with nasal polyps.

So far, a direct relationship between the occurrence of polymorphisms of this receptor and the development of both seasonal and year-long allergies has not been described. In our study, PAV/PAV homozygotes seem to be more likely to develop year-long allergies, but this tendency was not observed for seasonal allergies. Due to the different molecular mechanisms of the immune response in allergies and the nasal mucosa protection properties of the *TAS2R38* bitter taste receptor, this relationship may be difficult to determine and certainly requires further research [44].

Our results allow for a preliminary assessment of the distributions of the above-mentioned polymorphisms of the *TAS2R38* gene in CRSwNP patients and people without CRS symptoms. However, it cannot be directly translated into the entire population. Perhaps multicenter studies would allow for population-based inference. Our study confirms that there are certainly some associations between *TAS2R38* receptor polymorphism statuses and CRSwNP, but there is still no sufficient evidence to suggest a causal role of TRs in this disease. We believe there is a need for further research in this field to gain a better understanding of CRSwNP diseases.

We believe that the limitation of our work is the lack of information on flow-up. Since CRSwNP patients included in the study were those qualified for the primary FESS, we do not provide information on recurrences and the need for secondary surgical procedures or systemic steroid treatment. In the continuation of our research, we will pay special attention to the follow-up of all patients from the study group, which should give us more information on the further course of the disease in each patient and the opportunity to estimate the significance of the polymorphisms in this context. We also believe that multicenter studies might be a way to develop a representative study group for population-based inference.

Genetic studies targeting bitter taste receptors among CRS patients are aimed not only at searching for the molecular causes of the disease, but also at creating new tools for prognosing the course of the disease and individualizing the treatment method for each patient. Genotyping bitter taste polymorphisms in patients to predict the probable course of the disease might be a key factor in patient management. More data from different populations is still needed to globally assess the possible diagnostic and therapeutic potential of these receptors in CRS, providing the possibility to directly translate the properties of receptors into an individualized treatment plan for each patient.

## 5. Conclusions

Our study found significant relationships between genetic data and some clinical features among CRSwNP patients. We confirmed that the PAV/PAV diplotypes might have some protective properties, and carrying the AVI haplotype might predispose to the development of CRSwNP.

## Figures and Tables

**Figure 1 biomedicines-12-00168-f001:**
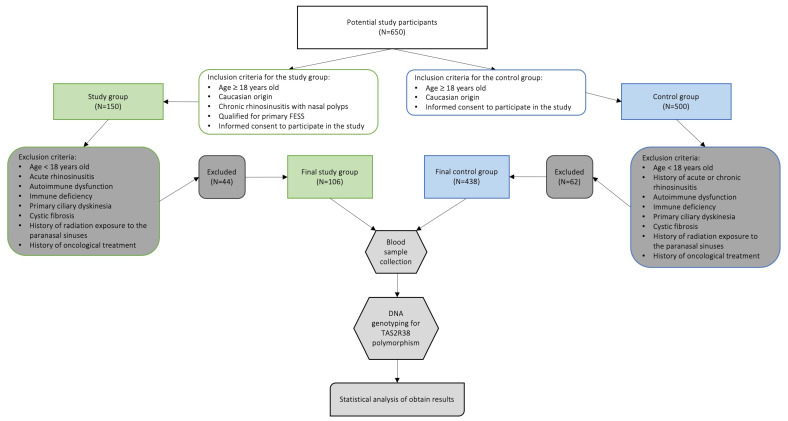
Study flowchart.

**Figure 2 biomedicines-12-00168-f002:**
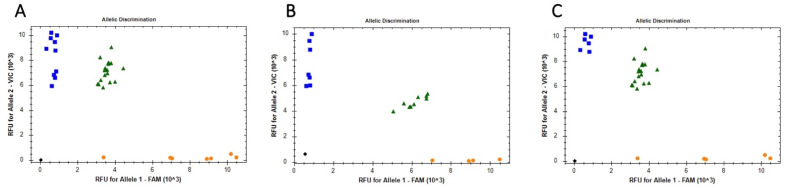
Allelic discrimination plot for (**A**) rs 1024939, (**B**) rs 713598 (**C**) rs 1726866 using TaqMan genotyping assays on 96 well plates. Blue squares and orange circles represent homozygous genotypes, whereas green triangles represent heterozygous genotypes. No template control (NTC) is marked as black diamond.

**Figure 3 biomedicines-12-00168-f003:**
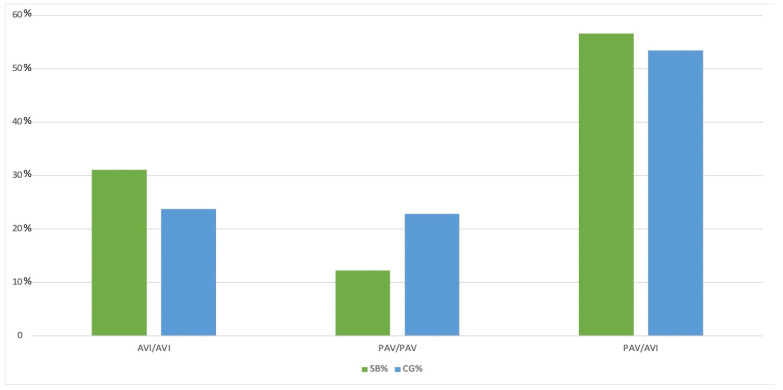
The *TAS2R38* diplotypes distribution in study group (N = 106) and control group (N = 438).

**Table 1 biomedicines-12-00168-t001:** Characteristics of the study group (N = 106).

		Study Group (N = 106)
Gender (%)	F	44 (41.51%)
	M	62 (58.49%)
Age (SD)		46.82 (14.02)
Weight (SD)		81.29 (17.15)
Height (SD)		172.89 (9.23)
BMI (SD)		27.12 (4.97)
PTC tasting (%)	0	36 (33.96%)
	1	70 (66.04%)
Age of the first symptoms (%)	<16 y.o. 16–40 y.o. >40 y.o.	23 (21.7%) 51 (48.11%) 32 (30.2%)
Asthma (%)	0 1	86 (81.13%) 20 (18.87%)
Seasonal allergies (%)		25 (23.58%)
Year-round allergies (%)		27 (25.47%)
Tabaco usage (%)		18 (16.98%)
VAS score (%)	0–3	47 (44.34%)
	4–7	35 (33.02%)
	8–10	24 (22.64%)
Median SNOT-22 score (SD)		32 (9.037)
Median Lund–Kennedy score (SD)		3 (1.14)
Median Lund–Mackay score (SD)		9 (5.474)

F = female, M = male, BMI = body mass index, PTC = phenylthiocarbamide, SNOT-22 = Sino-nasal Outcome Test-22, VAS = visual analog scale, SD = standard deviation, y.o. = years old.

**Table 2 biomedicines-12-00168-t002:** Distribution of haplotypes of the *TAS2R38* bitter taste receptor gene polymorphism in patients with CRSwNP (N = 106) and in the control group (N = 438).

			SG vs. CG	
Haplotype	Control Group N (%)	Study Group N (%)	OR (95% CI)	*p*-Value
PAV/PAV	100 (22.8)	13 (12.3)	1.0 *	
AVI/AVI	104 (23.7)	33 (31.1)	2.44 (1.21–4.90)	*p* = 0.01
PAV/AVI	234 (53.5)	60 (56.6)	1.97 (1.03–3.75)	*p* = 0.04
AVI/AVI; PAV/AVI	338 (77)	93 (88)	2.11 (1.136–3.941)	*p* = 0.02
AVI	443 (50.6)	126 (59.4)	1.43 (1.05–1.94)	
PAV	433 (49.4)	86 (40.6)	1.0 *	*p* = 0.02

* = reference gene; SG = study group, CG = control group.

**Table 3 biomedicines-12-00168-t003:** The dependency between diplotypes (PAV/AVI, AVI/AVI, PAV/PAV) and clinical features in study group (N = 106).

Value		PAV/AVI	AVI/AVI	PAV/PAV	*p*-Value
VAS	0–3	26 (55.32%)	12 (25.53%)	9 (19.15%)	0.3528
	4–7	20 (57.14%)	12 (34.29%)	3 (8.57%)	
	8–10	14 (58.33%)	9 (37.50%)	1 (4.17%)	
SNOT-22	Gr1	35 (58.33%)	17 (28.33%)	8 (13.33%)	0.7596
	Gr2	25 (54.35%)	16 (34.78%)	5 (10.87%)	
Lund–Kennedy	Gr1	42 (56.76%)	21 (28.38%)	11 (14.86%)	0.3772
	Gr2	18 (56.25%)	12 (37.50%)	2 (6.25%)	
Lund–Mackay	Gr1	37 (64.91%)	11 (19.30%)	9 (15.79%)	0.0158
	Gr2	23 (46.94%)	22 (44.90%)	4 (8.70%)	
PTC	0	24 (66.67%)	7 (19.44%)	5 (13.89%)	0.1751
	1	36 (51.43%)	26 (37.14%)	8 (11.43%)	
Asthma	0	52 (60.47%)	23 (16.74%)	11 (12.79%)	0.1273
	1	8 (40.00%)	10 (50.00%)	2 (10.00%)	
Seasonal allergies	0	46 (56.79%)	26 (32.10%)	9 (11.11%)	0.7876
	1	14 (56.00%)	7 (28.00%)	4 (16.00%)	
Year-round allergies	0	47 (59.49%)	26 (32.91%)	6 (7.59%)	0.0431
	1	13 (48.15%)	7 (25.93%)	7 (25.93%)	

VAS = visual analog scale, SNOT-22 = Sino-nasal Outcome Test-22, PTC = phenylthiocarbamide, Gr1—group with score under the median, Gr2—group with score over the median.

## Data Availability

Data available upon request.

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
