# Peer review of "TAS2R38 Bitter Taste Receptor Polymorphisms in Patients with Chronic Rhinosinusitis with Nasal Polyps Preliminary Data in Polish Population"

_biomedicines, 2024, doi:10.3390/biomedicines12010168_

Round 1
Reviewer 1 Report
Comments and Suggestions for Authors
The authors have presented the impact of SNP's of bitter taste receptors on clinical features of patients with CRS with nasal polyps
The study is well done and well-presented
Some issues need to be clarified
How were controls recruited? From the hospital or the community?
How was bias avoided in the selection of both cases and controls?
Please mention details of the sample size calculation and the study power
What tests were conducted in the control population to confirm that they do not have any relevant disease?
In Table 2: Please retain two decimals or a maximum of three decimals
example: p=0.01
Figure 2: Please mention the units of the Y-axis. Is it %? retain only two digits for the % 50 or 60 and not 50,00 or 60,00.
Line 307: Polish population and not polpopulation
Write a paragraph on the strengths and limitations of the study before the conclusions
Comments on the Quality of English LanguageA few spelling and grammatical errors need to be rectified
Author Response
Dear Reviewer,
Thank you very much for your thoughtful comments and efforts toward improving our manuscript.
We would like to address your concerns and comments point by point:
Reviewer: How were controls recruited? From the hospital or the community?
Response: The control group consisted of patients hospitalized in our Department for reasons other than CRSwNP and medical students who met the inclusion criteria and agreed to take part in the study. They all singed informed consent.
Reviewer: How was bias avoided in the selection of both cases and controls?
Response: The selection both in study and control group was no bias. All patients in the study group were qualified for planed primary FESS according to EPOS guidelines due to CRSwNP. We made every effort to ensure that both the control and study group was as representative as possible by inviting a similar number of men and women of different ages to participate in the study.
Reviewer: Please mention details of the sample size calculation and the study power
Response: The size of the study group was determined by the number of available patients and the restrictive inclusion criteria (CRSwNP prior to the first surgical intervention) and the timing of the pandemic with the lack of elective surgery, as well as costly gene testing, resulted in some limitations to the size of the study group. In our study, we used the results of estimating the power of the test as a function of effect size and sample size (On the Estimation of Power and Simple Size in Test of Independence, Asian Journal of Mathematics&Statistics, 2010,3:139-146; Table 3).
In our study, the power of the test was not high, which was influenced by the size of the study group of CRSwNP patients limited by the availability of patients without previous surgical interventions. Therefore, our conclusions are balanced, and the study should be continued to increase the power of the test.
Reviewer: What tests were conducted in the control population to confirm that they do not have any relevant disease?
Response: All participants in the control group underwent full laryngological examination. Patients included in the study, hospitalized due to planned surgery for example: ossicuplasty, explorant tympanotomy, benign laryngeal lesions had all tests necessary to qualify for general anesthesia, such as: morphology, coagulogram, chest x-ray and ECG. Medical students included in the study had no history of acute or chronic disease.
Reviewer: In Table 2: Please retain two decimals or a maximum of three decimals
example: p=0.01
Response: Thank you very much for this comment. In was corrected in the manuscript according to the reviewer's suggestion.
Reviewer: Figure 2: Please mention the units of the Y-axis. Is it %? retain only two digits for the % 50 or 60 and not 50,00 or 60,00.
Response: Thank you very much for this suggestion. In was corrected in the manuscript according to the reviewer's suggestion.
Reviewer: Line 307: Polish population and not polpopulation
Response: Thank you for this comment. It was corrected in the manuscript.
Reviewer: Write a paragraph on the strengths and limitations of the study before the conclusions
Response: Thank you very much for this comment. Limitations and strengths paragraph was added at the end of Discussion.
Added to the manuscript: We believe that the limitation of our work is in the lack of information on follow-up. Because CRSwNP patients included in the study were those qualified for the primary FESS, we do not provide information on recurrences and the need for secondary surgical procedures or systemic steroid treatment. This would give a better view on the course of the disease. We are monitoring those patients and we will include information on follow-up in further research.
Reviewer: A few spelling and grammatical errors need to be rectified
Response: Thank you for this value comment. Manuscript was carefully revised for linguistic and grammar correctness by English native speaker.
Reviewer 2 Report
Comments and Suggestions for Authors
Authors undoubtedly touched very interesting and significant for society health problem; thus, the aim is adequate for the research. However, I would like to ask the authors to sign out more the significance/connection between the bitter taste receptors and CRS with nasal polyps, and haplotypes because this is not completely revealed in the Introduction part.
Title has to include the “studies or pilot studies of the Polish population” as there are evidence about the CRS differences in different populations, so, might be also differences in bitter taste receptors…
Abbreviations, also for the haplotypes, should be deciphered everywhere where they are used for the first time.
M+M
Please, include the info about the time period (from-to), when the research was done. Caucasian origin, -mentioned for the patients raises the question, how was it detected? Or Caucasian origin people living in the Poland? This should be clarified, please…. Almost nothing is said in Introduction about this aspect, so, please, enroll some few sentences about the significance of different haplotypes between different origin persons and significance to search Caucasian origin people living in the Poland… Please, give precise inclusion criteria. Also give, please, references for the methods used in 2.2 subsection.
! Please correct once more the English as some inaccuracy mistakes appear (for instance, Lithuania should be written instead of Lithuana…)
The visual classification indicates just the presence of polyps, but were they the primary polyps or recurrent polyps? As this might be important, please describe how this was clarified?
I would like to ask the authors to develop a flowchart for the research design what will help to understand the whole M+M section better.
Please, add the Limitation paragraph at the end of discussion. Also, further directions of the research would be helpful to understand the future consequences.
Conclusions. Only one sentence “Our study found significant relationships between genetic data and some clinical features among CRSwNP patients” relates to the conclusion and even then, this is not elaborated and superficial. Please, remove all extra sentences, words, overwrite the conclusions by giving the basis of your findings for each conclusion.
Also, I would like to invite the authors to point out the real novelty of the research, as this is not understandable yet.
There are 2 old references (out of 45), what actually do not fit nowadays manuscripts. I would like to ask you to remove or to replace these 2 references with more appropriate ones.
Comments on the Quality of English LanguagePlease, ask the English native speaking scientist to go through your manuscript. There are mistakes in places here and there, and it is not understandable, are they connected to the carelessness or language problems... Some minor changes are requested.
Author Response
Dear Reviewer,
Thank you very much for your effort and time taken to review our manuscript. We agree with all your comments and suggestions, and we have implemented all the suggested changes in the manuscript, below is the point-by-point reply to your comments:
Reviewer: However, I would like to ask the authors to sign out more the significance/connection between the bitter taste receptors and CRS with nasal polyps, and haplotypes because this is not completely revealed in the Introduction part.
Response: Thank you very much for this comment. In was corrected in the manuscript according to the reviewer's suggestion.
Added to the manuscript: “Supertasters” seem to less likely develop chronic rhinosinusitis and in case of the disease have less severe symptoms and fever recurrence. CRS patients with AVI/AVI genotype seem more susceptible to severe course of the disease and more often require surgical treatment [12-14].
Reviewer: Title has to include the “studies or pilot studies of the Polish population” as there are evidence about the CRS differences in different populations, so, might be also differences in bitter taste receptors…
Response: Thank you very much for this comment. In was corrected in the manuscript according to the reviewer's suggestion.
Reviewer: Abbreviations, also for the haplotypes, should be deciphered everywhere where they are used for the first time.
Response: Thank you very much for this comment. In was corrected in the manuscript according to the reviewer's suggestion.
Reviewer: Please, include the info about the time period (from-to), when the research was done.
Response: Thank you for this comment. We included the study time period in the manuscript. Due to the fact that patients participating in the study were those qualified for planned surgery, during the SARS-CoV-2 pandemic, when all planned surgeries were canceled, we had to postpone the material collection.
Reviewer: Caucasian origin, -mentioned for the patients raises the question, how was it detected? Or Caucasian origin people living in the Poland? This should be clarified, please…. Almost nothing is said in Introduction about this aspect, so, please, enroll some few sentences about the significance of different haplotypes between different origin persons and significance to search Caucasian origin people living in the Poland… Please, give precise inclusion criteria.
Response: Thank you for this important comment. According to the data provided by Polish Central Statistical Office and EU Eurostat, polish population is one of the most homogeneous populations with 98% of people with Caucasian origin. The physical features of the participants combined with information about nationality and Polish roots were determining the Caucasian origin in our study. In the manuscript we cite: Risso, D. S., Mezzavilla, M., Pagani, L., Robino, A., Morini, G., Tofanelli, S., Carrai, M., Campa, D., Barale, R., Caradonna, F., Gasparini, P., Luiselli, D., Wooding, S., Drayna, D. Global diversity in the TAS2R38 bitter taste receptor: revisiting a classic evolutionary PROPosal. Sci Rep 2016, 6: 25506. https://doi.org/10.1038/srep25506 in which the differences in the global distribution of TAS2R38 polimorphisms are described in detail. We added this information to the Introduction paragraph.
Reviewer: Also give, please, references for the methods used in 2.2 subsection.
Response: Corrected in the manuscript according to the reviewer's suggestion.
Reviewer: Please correct once more the English as some inaccuracy mistakes appear (for instance, Lithuania should be written instead of Lithuana…)
Response: Thank you for this value comment. Manuscript was carefully revised for linguistic and grammar correctness.
Reviewer: The visual classification indicates just the presence of polyps, but were they the primary polyps or recurrent polyps? As this might be important, please describe how this was clarified?
Response: Thank you for this comment. All patients included in the study were patients qualified for the primary FESS. Before the procedure, they were all treated conservatively according to EPOS guidelines.
Reviewer: I would like to ask the authors to develop a flowchart for the research design what will help to understand the whole M+M section better.
Response: Thank you for this suggestion. A flowchart was added to the manuscript.
Reviewer: Please, add the Limitation paragraph at the end of discussion.
Response: Thank you very much for this comment. Limitation paragraph was added at the end of Discussion.
Added to the manuscript: We believe that the limitation of our work is in the lack of information on flow-up. Because CRSwNP patients included in the study were those qualified for the primary FESS, we do not provide information on recurrences and the need for secondary surgical procedures or systemic steroid treatment. This would give a better view on the course of the disease. We are monitoring those patients and we will include information on follow-up in further research.
Reviewer: Also, further directions of the research would be helpful to understand the future consequences.
Response: Thank you very much for this comment. In was corrected in the manuscript according to the reviewer's suggestion.
Added to the manuscript: In the continuation of our research, we will pay special attention on the follow-up of all patients from study group which should give us more information on the further course of the disease in each patient and the opportunity to estimate the significance of the polymorphisms in this context. We also believe that multicenter studies might be a way to develop a representative study group for population-based inference.
Reviewer: Conclusions. Only one sentence “Our study found significant relationships between genetic data and some clinical features among CRSwNP patients” relates to the conclusion and even then, this is not elaborated and superficial. Please, remove all extra sentences, words, overwrite the conclusions by giving the basis of your findings for each conclusion. Also, I would like to invite the authors to point out the real novelty of the research, as this is not understandable yet.
Response: Thank you very much for this comment. In was corrected in the manuscript according to the reviewer's suggestion.
Reviewer: There are 2 old references (out of 45), what actually do not fit nowadays manuscripts. I would like to ask you to remove or to replace these 2 references with more appropriate ones.
The reference 17: HARRIS, H., KALMUS, H. Genetical differences in taste sensitivity to phenylthiourea and to anti-thyroid substances. Nature 1949, 163(4153), 878. https://doi.org/10.1038/163878b0
Was changed to: Kim, U. K., Drayna, D. Genetics of individual differences in bitter taste perception: lessons from the PTC gene. Clinical genetics, 2005; 67(4), 275–280. https://doi.org/10.1111/j.1399-0004.2004.00361.x
The other old reference: Lund, V. J., Kennedy, D. W. Staging for rhinosinusitis. Otolaryngol Head Neck Surg. 1997, 117(3 Pt 2), S35–S40. https://doi.org/10.1016/S0194-59989770005-6, is a reference to the original work in which the Lund Kennedy original scale was published and we would like to maintain this reference.
Reviewer: Please, ask the English native speaking scientist to go through your manuscript. There are mistakes in places here and there, and it is not understandable, are they connected to the carelessness or language problems... Some minor changes are requested.
Response: Thank you for this value comment. Manuscript was carefully revised for linguistic and grammar correctness by English native speaker.
Round 2
Reviewer 2 Report
Comments and Suggestions for Authors
Dear Authors,
Thanks, almost all is OK. But not the Conclusions, - please, - remove these sentences "In the continuation of our research, we will pay special attention on the follow-up of all patients from study group which should give us more information on the further course of the disease in each patient and the opportunity to estimate the significance of the polymorphisms in this context. We also believe that multicenter studies might be a way to develop a representative study group for population-based inference." These are not conclusions from the results, they nicely will fit at the end of Line 333...
Otherwise, no more objections! Have a great New Year!
Author Response
Dear Reviewer!
Thank you very much for your positive assessment of our work and the improvements made. We agree with the last recommendation. We have moved the mentioned sentences to the limitaitons thus shortening the conclusions.